

# Integrated analysis of ischemic stroke datasets revealed sex and age difference in anti-stroke targets

Wen-Xing Li[1,2,*], Shao-Xing Dai[2,3,*], Qian Wang[2,3,*], Yi-Cheng Guo[2], Yi Hong[4], Jun-Juan Zheng[2,3], Jia-Qian Liu[2,3], Dahai Liu[4], Gong-Hua Li[2,3] and Jing-Fei Huang[2,3,5,6]

[1] Institute of Health Sciences, Anhui University, Hefei, Anhui, China
[2] State Key Laboratory of Genetic Resources and Evolution, Kunming Institute of Zoology, Chinese Academy of Sciences, Kunming, Yunnan, China
[3] Kunming College of Life Science, University of Chinese Academy of Sciences, Kunming, Yunnan, China
[4] Center for Stem Cell and Translational Medicine, School of Life Sciences, Anhui University, Hefei, Anhui, China
[5] KIZ-SU Joint Laboratory of Animal Models and Drug Development, College of Pharmaceutical Sciences, Soochow University, Kunming, Yunnan, China
[6] Collaborative Innovation Center for Natural Products and Biological Drugs of Yunnan, Kunming, Yunnan, China
* These authors contributed equally to this work.

Corresponding authors
Dahai Liu, seansean2014@126.com
Gong-Hua Li, ligonghua@mail.kiz.ac.cn
Jing-Fei Huang, huangjf@mail.kiz.ac.cn

## ABSTRACT

Ischemic stroke is a common neurological disorder and the burden in the world is growing. This study aims to explore the effect of sex and age difference on ischemic stroke using integrated microarray datasets. The results showed a dramatic difference in whole gene expression profiles and influenced pathways between males and females, and also in the old and young individuals. Furthermore, compared with old males, old female patients showed more serious biological function damage. However, females showed less affected pathways than males in young subjects. Functional interaction networks showed these differential expression genes were mostly related to immune and inflammation-related functions. In addition, we found ARG1 and MMP9 were up-regulated in total and all subgroups. Importantly, IL1A, ILAB, IL6 and TNF and other anti-stroke target genes were up-regulated in males. However, these anti-stroke target genes showed low expression in females. This study found huge sex and age differences in ischemic stroke especially the opposite expression of anti-stroke target genes. Future studies are needed to uncover these pathological mechanisms, and to take appropriate pre-prevention, treatment and rehabilitation measures.

## INTRODUCTION

Stroke is a common neurological disorder which has become the second leading cause of death worldwide (*Feigin et al., 2014*). It is predicted that by 2030, there could be almost 12 million stroke deaths, 70 million stroke survivors, and more than 200 million disability-adjusted life-years (DALYs) lost from stroke each year (*Feigin et al., 2014*). The

major epidemiological risk factor for stroke is hypertension. Researchers showed that even mild hypertension is strongly associated with an increased incidence of stroke, and severe hypertension will greatly increase the burden of stroke (*Kim, Cahill & Cheng, 2015*). In addition to hypertension, other major risk factors such as diabetes mellitus, high levels of cholesterol and triglycerides, obesity, and smoking also contribute to stroke (*Kim, Cahill & Cheng, 2015*; *Liu et al., 2011*). Stroke burden in high-income countries is very serious, and the burden of stroke increases rapidly in low-income and middle-income countries in recent years with the rapid development of social economy (*Feigin et al., 2014*; *Kim, Cahill & Cheng, 2015*).

Ischemic stroke specifically refers to central nervous system (CNS) infarction accompanied by overt symptoms, silent CNS infarction causes no known symptoms (*Sacco et al., 2013*). Ischemic strokes account for the vast majority (85%) of stroke events vs. 15% for hemorrhagic strokes (*Musuka et al., 2015*). It is reported that the incidence, prevalence, morbidity and mortality of ischemic stroke were influenced by sex (*Barker-Collo et al., 2015*; *Reeves et al., 2008*). Furthermore, the stroke preventive care, clinical characteristics at stroke onset, pre-hospital and in-hospital delays, diagnostic and treatment procedures, drug efficacy and response, rehabilitation and post-stroke recovery were also affected by sexual dimorphism (*Reeves et al., 2008*). A study of Finland and Sweden population suggested males had a higher incidence of ischemic stroke and acute coronary heart disease (CHD) than females (*Hyvarinen et al., 2010*). However, *Tomita et al. (2015)* showed that female is a risk factor for stroke severity and unfavorable functional outcome in patients with cardioembolic stroke.

It is well known that the severity of stroke increased with the growing of age. Recently, a meta-analysis showed a periprocedural hazard ratio (HR) of 2.16 for stroke and death in patients aged 65–69 years compared with patients <60 years, and with higher HRs of about 4.0 for patients more than 70 years for patients assigned to carotid artery stenting (CAS) (*Howard et al., 2016*). And the outcomes of female/male mortality ratios for stroke were also had differences in the stratified age group (*Reeves et al., 2008*). A Japanese study including 33,953 patients showed women are more frequently had ischemic stroke, hypertension, dyslipidemia and other cardioembolic events with the age adjustment (*Maeda et al., 2013*). It is noteworthy that the age of this study population was relatively older (Female vs. Male: $75.0 \pm 11.7$ vs. $69.3 \pm 11.4$ years). A previous review showed that premenopausal women are at a lower risk of stroke compared to men in the same age. However, the incidence of ischemic stroke increases rapidly in the postmenopausal women (*Appelros, Stegmayr & Terent, 2009*). Therefore, the interpretation of sex difference on ischemic stroke should take into account patients' age.

However, the mechanism of this apparent sex difference in ischemic stroke risk is not fully understood. A previous study suggested that there was a huge sexually dimorphic in immune cell gene expression profiles following cardioembolic stroke, and this difference becomes maximum at 24 h after stroke (*Stamova et al., 2014*). To provide an overall and clearer picture on the sex and age difference of ischemic stroke, we integrated microarray data from two platforms, explored these differences from whole gene expression profiles,

metabolic pathways, functional interactions and anti-stroke targets. We also investigated the outcomes of sex difference in old and young patients.

## METHODS

### Microarray data collection

Human ischemic stroke microarray datasets were searched and downloaded from NCBI-GEO database (http://www.ncbi.nlm.nih.gov/geo) and EMBL-EBI ArrayExpress database (https://www.ebi.ac.uk/arrayexpress/) with the keywords of "Ischemic Stroke" in January 2016. The data selection criteria were: (1) all datasets were genome-wide; (2) the samples of each dataset must include ischemic stroke patients and controls; (3) the number of cases and controls in each dataset must ≥3; (4) non cell line samples and (5) raw data or normalized expression matrix was available. Based on the above criteria, were finally chose two datasets for our integrate analysis (GSE16561 and GSE22255). In the dataset of GSE16561 (contributed by Barr TL), a total of 63 samples (including 39 patients and 24 controls) were tested using the Illumina HumanRef-8 v3.0 Expression BeadChip. And the GSE22255 dataset (contributed by Krug T) included 20 patients and 20 controls and each sample was hybridized to Affymetrix Human Genome U133 Plus 2.0 Array. All mRNA samples were from peripheral blood mononuclear cells (PBMCs) (*Barr et al., 2010a*; *Krug et al., 2012*). Details of the two datasets appear in Table S1. Neither of two datasets have information about disease history (such as hypertension) or stroke subtype. We divided the samples in these datasets into old (age ≥ 60) and young (age < 60) groups. The age distribution of ischemic stroke patients and controls appear in Table S2. There was no difference in age between male and female both in patients and controls (Table S2).

### Data preprocessing

R v3.2.2 was performing data preprocessing. We used Robust Multichip Average (RMA) algorithm in oligo package (*Carvalho & Irizarry, 2010*) to normalize the raw expression data and generate normalized gene expression intensity of GSE22255. Because the GSE16561 dataset provided a RMA normalized gene expression matrix (*Barr et al., 2010a*), therefore, we download the matrix in NCBI-GEO. Gene annotation, integration and renormalization of the two datasets were carried out using custom written Python code (Supplemental Information 1). We have removed probes that had no gene annotation or that matched multiple gene symbols. Next, we calculated the average expression value of multiple probe IDs that matched to an official gene symbol, and took this value to represent the expression intensity of the corresponding gene symbol. The renormalization method was reported in our previous publication (*Li et al., 2016*). The distributions of RMA processed and global renormalized gene expression values in two studies were showed in Fig. S1.

### Differential expression analysis

Differential expression genes analysis was using limma package (*Ritchie et al., 2015*) in R v3.2.2. The empirical Bayes algorithm (function "eBayes") was used to detect differentially expressed genes between ischemic stroke patients and controls. Differentially expressed genes were defined as the absolute value of logarithmic transformed fold-change

$(\log(FC)) \geq \log2(1.5)$ and a $P$ value $\leq 0.05$. Up- and down-regulated genes were considered as $\log(FC) \geq \log2(1.5)$ and $\log(FC) \leq \log2(1.5)$, respectively. Differentially expressed genes in male, female, old and young group were calculated as male patients vs. male controls, female patients vs. female controls, old patients vs. old controls and young patients vs. young controls. In order to verify the sex and age differences of gene expression profiles in ischemic stroke were not merely related to male vs. female or age changes in gene expression, we calculated the genes that differentially expressed in males vs. females and olds vs. youngs in total group. To explain the sex and age difference in ischemic stroke, we used Venn diagram to show the number of differentially expressed genes in male vs. female, old vs. young and each cases-controls groups. And we also used the Venn diagram to show the number of up- and down-regulated genes in male and female group, and old and young group.

## KEGG pathways analysis

We used Functional Annotation Tool in DAVID Bioinformatics Resources 6.7 (*Huang da, Sherman & Lempicki, 2009*) to perform KEGG (Kyoto Encyclopedia of Genes and Genomes) pathway enrichment analysis and a pathway with a $P$ value $\leq 0.01$ was considered to be significantly enriched. We got 5, 6, 3, 5 and 1 significant enriched pathways in total, male, female, old and young group, respectively. Then we took the union set of enriched pathways in male and female, and old and young group, and showed the pathways and related gene expression profiles.

To explore the role of age in sex differences in ischemic stroke, we classified the datasets into four groups: old male, old female, young male and young female. Each dataset included 31 (20 patients and 11 controls), 32 (21 patients and 11 controls), 16 (7 patients and 9 controls) and 24 (11 patients and 13 controls) samples, respectively. We also performed KEGG enrichment analysis in each group and focused on the difference in the pathway level and gene expression profiles in enriched pathways between male and female in the different age group.

## Functional interaction analysis

The GeneMANIA prediction server (http://www.genemania.org/) (*Zuberi et al., 2013*) was used to perform functional interaction analysis. We used the differentially expressed genes list in total, male, female, old and young group as input parameters, respectively. Since many reports were based on the total ischemic patients, we analysed the total group as a reference. The server can find other genes that are related to the set of input genes and produce a functional association network based on their relationships, such as pathways, co-expression, co-localization, genetic interaction, physical interaction and so on. We chose the relationships of physical and genetic interactions, and pathway to produce function cluster analysis.

## Anti-stroke targets analysis

Anti-stroke target genes were searched and downloaded from Thomson Reuters Integrity Database (https://integrity.thomson-pharma.com/integrity). We got a total of 531 anti-stroke target genes and then mapped to our screened differentially expressed genes in total and subgroups (male, female, old and young group). The total group was shown as the

reference. Then we compared the difference of target genes between male and females, and old and young patients and used "barplot" function to show the results.

### Anti-stroke targets validation

Validation of the above differentially expressed anti-stroke target genes was conducted in an independent patient cohort. Through our rigorous screening, we finally chose GSE37587 dataset (contributed by Barr TL) for validation. In this dataset, 34 baseline ischemic stroke mRNA samples were from peripheral blood mononuclear cells (PBMCs), and were tested using the Illumina HumanRef-8 v3.0 Expression BeadChip (Table S1). We downloaded the RMA normalized gene expression matrix of this dataset in NCBI-GEO and preprocessed using our custom Python script (Supplemental Information 1). This dataset does not have controls, so we used controls in the GSE16561 and GSE22255 datasets to validate the differentially expressed anti-stroke target genes in total and four subgroups.

## RESULTS

### Overview of differentially expressed genes

Venn diagram of the up- and down-regulated genes showed in Fig. 1. There were 55 up- and 6 down-regulated genes in the total group. In the subgroup analysis by sex, we have detected 140 up- and 18 down-regulated genes in the male group, and 150 up- and 40 down-regulated genes in the female group, respectively (Fig. 1A). Furthermore, we found 38 up- and 59 down-regulated genes in the old group, and 51 up- and 1 down-regulated genes in the young group (Fig. 1B). There were only 8 genes over-expressed (ARG1, CA4, CKAP4, DUSP1, FOS, MMP9, ORM1 and RGS2) and 2 genes low- expressed (CCR7 and ID3) both in male and female group. Additionally, in old and young group only showed 6 overlapped over-expressed genes (ARG1, CA1, FAM46C, MMP9, SDPR and THBS1). Differentially expressed genes validation showed in Fig. S2. We got 305 and 532 differentially expressed genes in male vs. female and old vs. young group. However, the numbers of overlapping genes in the two groups and other cases-controls groups only were 13 and 19. Therefore, these results suggested a huge difference in the global gene expression profiles between male and female, and also in old and young in ischemic stroke, and these differences in gene expression profiles were mainly caused by disease states.

### Sex and age difference in pathway view

Gene expression profiles of enrichment pathways in total group showed in Fig. S3. The Toll-like receptor signaling pathway, NOD-like receptor signaling pathway, Cytokine-cytokine receptor interaction and Chemokine signaling pathway were significantly enriched. The union enriched pathways in male and female, and old and young patients were 8 and 6, respectively. Figure 2 displayed the gene expression profiles of enrichment pathways in each subgroup. There were 9 up-regulated genes in NOD-like receptor signaling pathway in male patients, whereas female seemed unaffected in this pathway. However, female patients showed more influence in Systemic lupus erythematosus pathway than male (Fig. 2A). In Cytokine-cytokine receptor interaction pathway, male patients displayed more over-expressed genes whereas female showed more low-expressed genes. For old

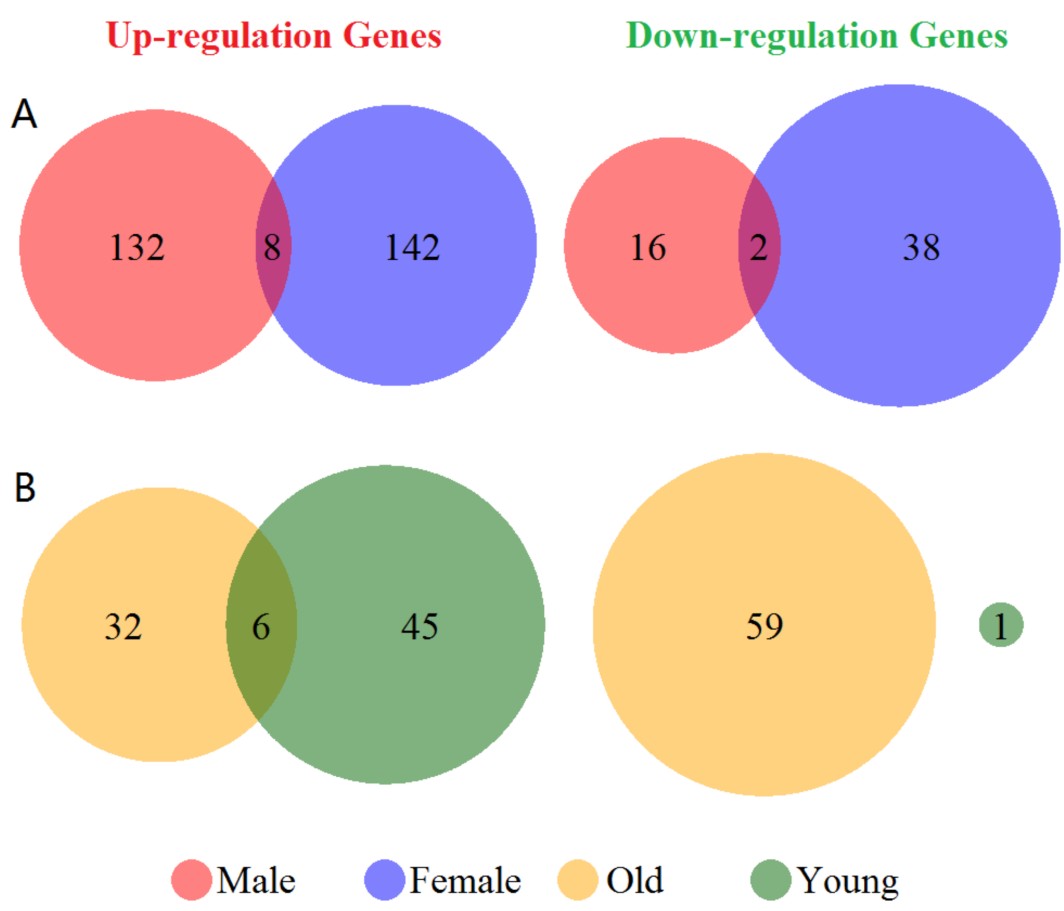

**Figure 1** **Venn diagram of the differentially expressed genes grouped by sex and age.** (A) Up- and down-regulated genes in male and female groups (red and blue cycle). (B) Up- and down-regulated genes in old and young groups (yellow and green cycle). The junction of the circle shows the overlapped genes in the two groups.

patients, the Hematopoietic cell lineage, Primary immunodeficiency, Antigen processing and presentation, Systemic lupus erythematosus and T cell receptor signaling pathway were severely affected. However, only NOD-like receptor signaling pathway was significantly affected in young patients (Fig. 2B).

We further analyzed sex difference in the ischemic stroke stratified by old and young patients (Fig. S4). The results showed that old female patients had multiple pathways seriously down-regulated (including Hematopoietic cell lineage, NOD-like receptor signaling pathway, Cytokine-cytokine receptor interaction, T cell receptor signaling pathway, Primary immunodeficiency and other pathways). However, old male patients showed the opposite results that several genes were over-expressed in the above pathways. Overall, old female showed more affected genes and pathways than old male (Fig. S4A). In the young group, we got an interesting result that males showed more up- and down-regulated genes than females in the enriched pathways (Fig. S4B). Certainly, these influenced genes and pathways were more serious in old patients than young patients both in male and female.

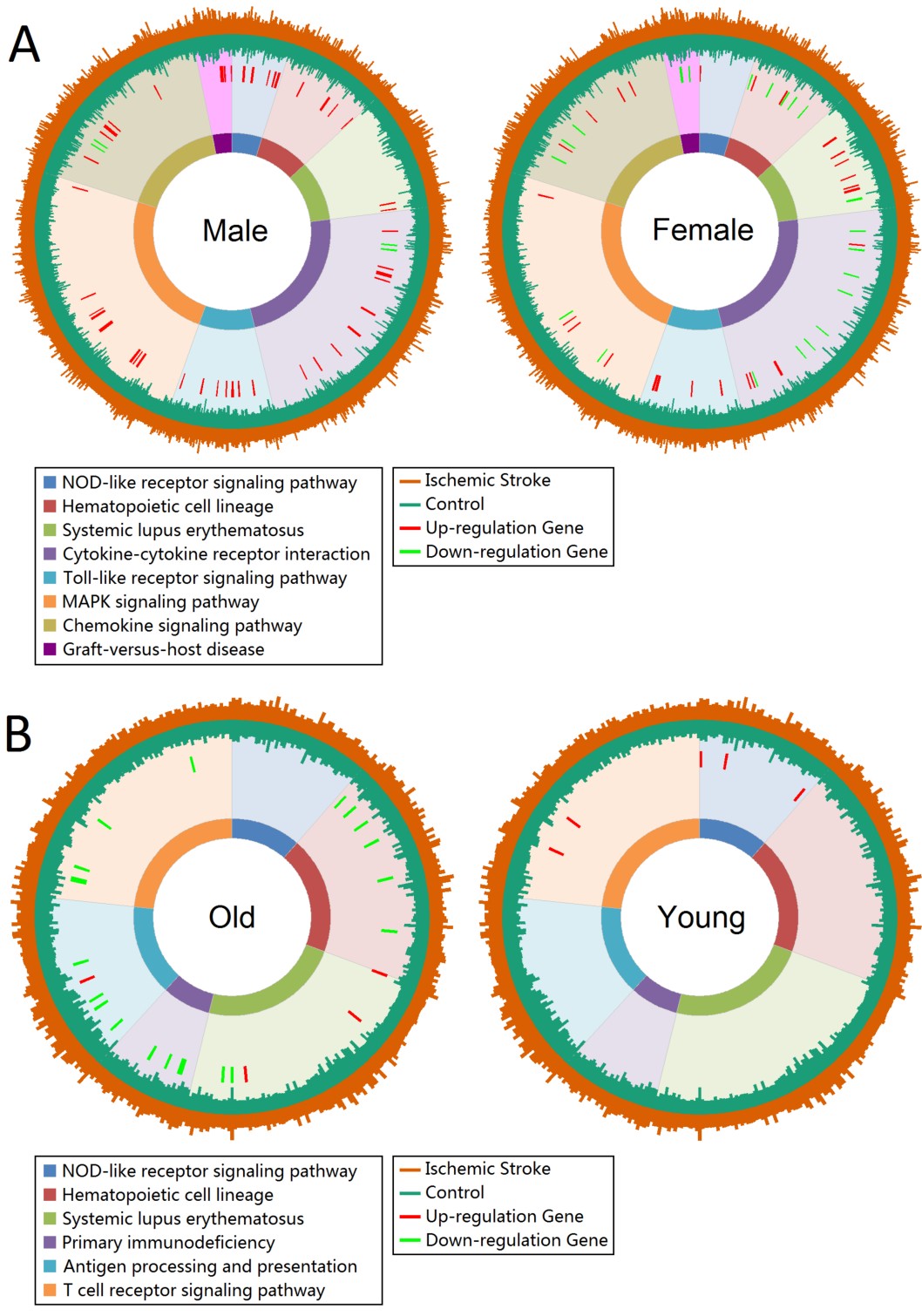

**Figure 2** **Gene expression profiles of enrichment pathways.** (A) Gene expression profiles in male and female groups. (B) Gene expression profiles in old and young groups. Pathways are represented by different colors. 

**Figure 2 (...continued)**
The length of the first layer of lines outside the circle represents the expression value in ischemic stroke patients and the length of the second layer of lines within the circle represents the expression value in controls. The up- and down-regulated genes are marked as red and green lines in the third layer.

## Function interaction network

Figures S5–S9 showed the function interaction network results in total, male, female, old and young group. The results showed a larger difference in the function interactions differentially expressed genes between male and female patients (Figs. S6–S7), and also in old and young patients (Figs. S8–S9). Table 1 showed top 5 enriched functions based on function interaction network in total and each subgroups. The enriched functions in total, male, female and old groups were mostly related to immune and inflammation-related functions. However, young patients showed more angiogenesis and vasculature development functions.

## Mapped anti-stroke targets in different groups

We mapped 5, 18, 5, 3 and 7 anti-stroke targets from differentially expressed genes in total, male, female, old and young groups, respectively. These targets were shown with the yellow cycle in functional interaction network (Figs. S5–S9). Figure 3 showed gene expression logFC of these targets in each group. MMP9 was over-expressed in the total and all subgroups (Figs. 3A, 3B and 3C). IL1A, IL1B, IL6 and IL8 were significantly up-regulated in male patients. However, these targets showed low-expressed in female patients. Furthermore, PDK4, HSPA1A and TLR4 were over-expressed in female patients whereas these targets were unaltered in male patients (Fig. 3B). In addition, targets of PDK4, HSPA1A, JUN, IL8, SOD2, JUNB and FOSB showed a different expression status in old and young patients (Fig. 3C).

## Validation of anti-stroke targets

Validation of anti-stroke target genes in the total and four subgroups showed in Fig. S10. Similar to the above findings, IL1A, IL1B, IL6, IL8, TNF and other 10 anti-stroke target genes were significantly up-regulated in male patients. However, only PDK4 was over-expressed, other target genes had no difference in the female group. Further, JUN, IL8, SOD2, PTGS2, JUNB and FOSB were up-regulated in the old group and these targets showed no difference in the young group.

## DISCUSSION

The present study showed a huge difference of gene expression profiles in male and female, and also in old and young ischemic stroke patients. These genes were mostly enriched in the immune-related pathways and cell signaling pathways. Furthermore, several anti-stroke targets showed the opposite expression between males and females.

Previous studies have made great effort on sex and age difference in ischemic stroke and got many valuable discoveries (*Howard et al., 2016*; *Stamova et al., 2014*; *Tian et al., 2012*). However, the sex heterogeneity of anti-stroke target genes expression still have no report. In the present study, we adopted a multi-platform integrated analysis method

**Table 1  Top five enriched functions based on functional interaction network in each group.**

| Functions | FDR | Genes in network | Genes in genome |
|---|---|---|---|
| *Total* | | | |
| Response to bacterium | 1.47E−10 | 14 | 166 |
| Defense response to bacterium | 6.97E−09 | 10 | 76 |
| Cell chemotaxis | 1.52E−08 | 12 | 157 |
| Positive regulation of defense response | 5.45E−08 | 13 | 228 |
| Defense response to other organism | 7.71E−08 | 12 | 188 |
| *Male* | | | |
| Inflammatory response | 2.43E−12 | 24 | 282 |
| Cell chemotaxis | 1.53E−07 | 15 | 157 |
| Negative regulation of multicellular organismal process | 1.60E−07 | 17 | 229 |
| Positive regulation of defense response | 1.60E−07 | 17 | 228 |
| Leukocyte migration | 1.60E−07 | 17 | 230 |
| *Female* | | | |
| Defense response to other organism | 1.57E−13 | 23 | 188 |
| Response to bacterium | 2.19E−10 | 19 | 166 |
| Defense response to bacterium | 1.10E−08 | 13 | 76 |
| Negative regulation of viral genome replication | 3.25E−08 | 10 | 38 |
| Secretory granule | 3.86E−08 | 16 | 152 |
| *Old* | | | |
| Defense response to bacterium | 5.43E−07 | 10 | 76 |
| Defense response to other organism | 1.02E−06 | 13 | 188 |
| Secretory granule | 1.24E−05 | 11 | 152 |
| Response to fungus | 1.55E−05 | 5 | 11 |
| Response to bacterium | 1.88E−05 | 11 | 166 |
| *Young* | | | |
| Angiogenesis | 2.57E−04 | 10 | 237 |
| Regulation of vasculature development | 4.42E−04 | 8 | 146 |
| Inflammatory response | 4.42E−04 | 10 | 282 |
| Regulation of angiogenesis | 2.28E−03 | 7 | 131 |
| Positive regulation of defense response | 4.58E−03 | 8 | 228 |

to conduct in deep analysis on sex and age difference in ischemic stroke especially for the anti-stroke target genes. We have also explored the interactions of sex and age on dysregulated pathways. Our study found there were more up- and down-regulated genes in females than males in this study. This difference was also found in *Stamova et al. (2014)*, which showed a more serious peripheral immune cell dysfunction in females than males at ≤3, 5 and 24 h following cardioembolic stroke. Furthermore, compared with young patients, old patients showed less up-regulated genes and more down-regulated genes. In total and subgroup analysis, we found ARG1 and MMP9 were significantly higher expression in patients than controls. Indicate that these two genes were strongly associated with the pathological process of ischemic stroke. ARG1 (Arginase 1) is an essential enzyme that converts L-arginine to L-ornithine in the urea cycle. Role of ARG1 in cardiovascular

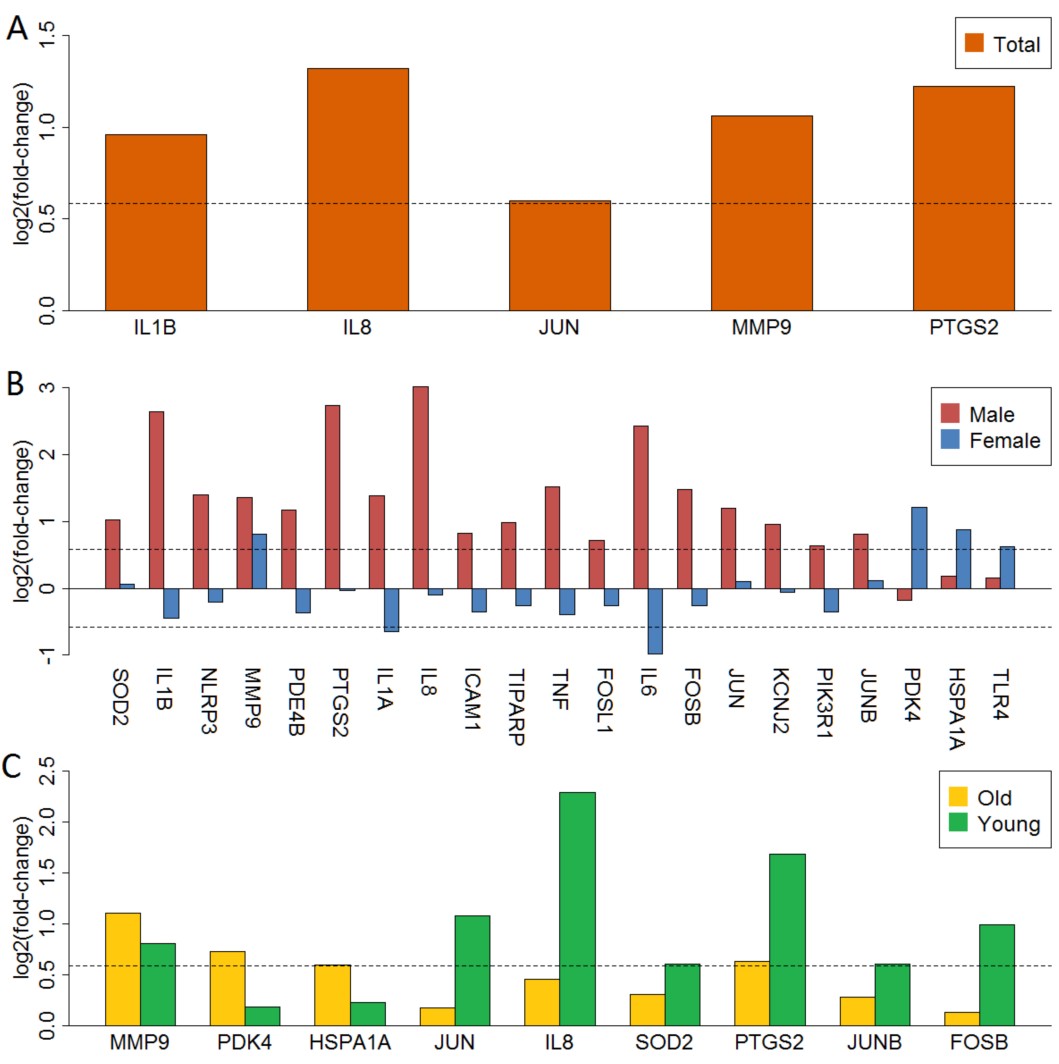

**Figure 3   LogFC barplot of mapped anti-stroke target genes.** (A) LogFC in total group. (B) LogFC in male and female groups. (C) LogFC in old and young group. The horizontal dashed lines represent the logFC cutoff of the up- and down-regulated genes.

disease has been studied extensively, especially in the regulation of the immune system. An increasing number of studies showed closely relations between ischemic stroke and systemic immune consequences (*Asano, Chantler & Barr, 2016*). The immunosuppressive effect of ARG1 was demonstrated by *El Kasmi et al. (2008)*, and suggested ARG1 inhibition dramatically improves host immune response to tuberculosis infection. A recent study showed that stroke-related immune suppression is associated with activated neutrophil and ARG1 release in middle cerebral artery occlusion (MCAO) mouse (*Sippel et al., 2015*). Furthermore, increased expression of ARG1 in microglia following ischemic stroke is linked to improved tissue remodeling and behavioral recovery (*Asano, Chantler & Barr, 2016*). MMP9 belongs to the matrix metalloproteinase family and was involved in the breakdown of extracellular matrix in normal physiological processes. The association between blood brain barrier (BBB) dysfunction and elevated MMP9 levels in ischemic patients was

demonstrated both in a mouse model (*Asahi et al., 2001*) and clinical studies (*Barr et al., 2010b*). In our study, patients with ischemic stroke showed severe immune function defects. Furthermore, in anti-stroke target analysis, we found MMP9 was up-regulated in the total and all subgroups. Taken together, these observations reinforce the view of severity immune and cerebrovascular dysfunction in ischemic stroke. Additionally, mRNA samples in the present study were all from PBMCs, thus we suggest that ARG1 and MMP9 could be used as biomarkers for the detection of ischemic stroke.

Altered gene expression is an important feature of ischemic stroke and affects proteins in numerous biological functions. This study showed different degrees of damage in NOD-like receptor signaling pathway, Hematopoietic cell lineage, Systemic lupus erythematosus and Cytokine-cytokine receptor interaction in males and females. After considering the age, more biological impairment emerged both in males and females. Old females present a large number of down-regulated genes in various pathways. However, old males only showed a handful of down-regulated genes in the affected pathways (Fig. S4). The damage pathways of NOD-like receptor signaling pathway, Hematopoietic cell lineage and Cytokinecytokine receptor interaction were also found in middle cerebral artery occlusion induced ischemic stroke mice (*Quan et al., 2015*). Furthermore, a recent study showed affected Ribosome, Toll-like receptor signaling pathways, MAPK signaling pathways, and Chemokine signaling pathways in the ischemic rodent brain (*Liang et al., 2015*) and identified three key autophagy genes (STAT3, NFKB1, and RELA) use the method of constructing the autophagy-related pathway network (*Liang et al., 2015*). Our results showed a slight difference of Toll-like receptor signaling pathway, MAPK signaling pathway, and Chemokine signaling pathway between male and female (Fig. 2). However, when stratified by age, we observed more serious abnormal gene expression in old females than old males subjects (Fig. S4). One possible explanation of these phenomena was the role of sex hormones, for example the positive effects of estrogen on the cerebral circulation in females (*Appelros, Stegmayr & Terent, 2009*). Some studies have indeed observed the associations between estrogen levels and the stroke risk in females and suggest that post menopausal women have a high incidence of stroke than men (*Appelros, Stegmayr & Terent, 2009*; *Reeves et al., 2008*). However, randomized controlled trials among post menopausal women showed no advantages of exogenous oestrogen therapy, even increases the risk of stroke (*Reeves et al., 2008*). Our results found no difference in the estrogen-related pathways in females. These results indicate that the apparent sex difference in ischemic stroke may affect by other factors.

The results of functional interaction network showed the affected proteins were mostly enriched in immune and inflammatory responses functions in total, male, female and old groups. Previous studies showed several interleukins (IL1, IL6, IL8, TNF, etc.) play a crucial role in immune processes and strongly associated with ischemic stroke (*Kostulas et al., 1999*; *Tian et al., 2012*). Interleukin-1 (IL1), a highly active pro-inflammatory cytokine, acts as a main regulator of inflammation process and triggers a cascade of inflammatory mediators by activation of the IL1 receptor (*Dinarello, Simon & Van der Meer, 2012*). There are two related but distinct IL1 genes, IL1A and IL1B, encoding IL1$\alpha$ and IL1$\beta$, respectively. Experimental results show abnormal levels of

IL1$\alpha$ or IL1$\beta$ will lead to inflammatory diseases (*Dinarello, Simon & Van der Meer, 2012*). IL1 is also known to involve in the neurodegeneration of acute and chronic brain disorders, including ischemic stroke, Alzheimer's disease and Parkinson's disease, although the cytokine is thought to be involved in the recovery of neurological functions (*Dinarello, Simon & Van der Meer, 2012*; *Kostulas et al., 1999*). IL6 is a key early mediator of the inflammatory and overall immune response and plays an important role in the development of pathological conditions (*Ferrarese et al., 1999*; *Ridker et al., 2000*). And it's produced by many different cells including monocytes, macrophages, fibroblasts, endothelial cells, keratinocytes, mast cells, T-lymphocytes, and also by microglia and astrocytes (*Ferrarese et al., 1999*). An increasing number of experimental observations showed that IL6 plays a central role in the pathogenesis of several ischemic cardiovascular disorders, including ischemic stroke (*Quan et al., 2015*). A recent report suggested that IL6 is essential for the promoting effects of social interaction on the neurogenesis as well as long-term functional recovery after ischemic stroke (*Meng et al., 2015*). TNF belongs to the tumor necrosis factor family and tumor necrosis factor-$\alpha$ (TNF-$\alpha$) is the main member that has many functions including inflammation, sepsis, lipid and protein metabolism, haematopoiesis, angiogenesis and host resistance to parasites and malignancy (*Beutler & Cerami, 1988*; *Old, 1985*). A previous study demonstrated that TNF-$\alpha$ activates the expression of pro-adhesive molecules on the endothelium, which causes the leukocyte accumulation, adherence, and migration from capillaries into the brain (*Akira et al., 1990*). Furthermore, TNF-$\alpha$ also induces the expression of IL1, IL6 and other cytokines. In the acute phase of ischemia, TNF-$\alpha$ and IL1B as inflammatory factors, cause the acceleration of inflammatory lesions, and induce cell necrosis or apoptosis (*Kawai, 1999*).

In anti-stroke target analysis of the total group, the results showed that IL1B, IL8, JUN, MMP9 and PTGS2 were significantly up-regulated (Fig. 3A). In addition, IL1A, IL6, TNF and other anti-stroke targets were also presented an up-regulated trend. These findings were consistent with previous reports (*Kostulas et al., 1999*; *Ridker et al., 2000*). However, after sex stratification we found that IL1A, IL1B, IL6, IL8, TNF and other 13 anti-stroke targets were all up-regulated in male patients. Whereas most of these anti-stroke targets showed no difference or opposite expression in female patients, especially IL1A and IL6. The change ranges of these anti-stroke targets in females apparently lower than males. Several pro-inflammatory cytokines, such as IL1, IL6 and TNF have been identified as promising therapeutic targets in immune and inflammatory related diseases. Extracellular acidification inhibits IL1$\beta$ production showed improved cell environment and reduce acidosis, suggested a protective effect of chronic neurodegenerative diseases (*Jin et al., 2014*). IL6 has been proved as a target for the treatment of several inflammatory diseases (*Metz et al., 2006*). Recently, several new IL1 and IL6 inhibitors were developed and their therapeutic effects have been successfully validated (*Divithotawela et al., 2016*; *Wiesinger et al., 2009*). Furthermore, our pathway enrichment results suggested that interferon signaling may be an important factor in sex-biased mechanisms. Previous studies also provided many evidences that interferon signaling genes were closely related to stroke (*Lin & Levison, 2009*; *Simmons et al., 2013*). Thus, we extracted 23 interferon signaling genes in our datasets. Then we used GeneMANIA to perform functional interaction analysis between interferon

signaling genes and differentially expressed anti-stroke target genes in each group. We found strong connections (including as co-expression, pathway, physical interactions, shared protein domains and co-localization) in many interferon signaling genes and anti-stroke genes. And the connection strength varies greatly in different groups (Figs. S11–S15).

Inflammation is an essential component of pathological mechanisms in charge of ischemic stroke. Brazilein is an important pro-inflammatory cytokines inhibitor, it has been demonstrated a neuroprotective effect though suppress TNF$\alpha$ and IL6 mRNA expressions, the treatment prospects on ischemic stroke have been widely reported (*Shen et al., 2007*; *Ye et al., 2006*). Activated p38 mitogen-activated protein kinase (MAPK) is closely related to cerebral ischemia disease, SB239063 is an ATP competitive N-substituted imidazole-based inhibitor. Previous studies showed it has a strong inhibitory effect of IL1$\beta$, IL6, IL10, TNF and other circulating cytokines, and suggested a potential therapeutic for stroke patients (*Bison et al., 2011*; *Zhang et al., 2015*). Furthermore, a previous study suggested a neuroprotective effect of benzylideneacetophenone derivatives in stroke models. These compounds act on the JAK/STAT and MAPK pathways, inhibit mRNA expressions of IL1$\beta$, IL6 and TNF (*Jang et al., 2009*). However, these anti-stroke targets showed conversely expression between males and females in the present study. Therefore, if researchers were not taking into account these sex differences in the investigation, the results may conceal the real difference. And the observed increase of target genes may simply be attributable to the differences in males. Based on our findings, we suggest that care be taken in the treatment of ischemic stroke using these cytokine inhibitors.

In conclusion, this study first demonstrated the sex and age differences in ischemic stroke using the integrated microarray datasets. We revealed the sex difference in ischemic stroke was influenced by age. More importantly, our results showed a conversely expression in multiple anti-stroke target genes between male and female. Some inhibitors of these targets have already been approved for treatment of neurological diseases, such as ischemic stroke. Therefore, in future drug development and clinical therapy stage of ischemic stroke, this sex difference should be seriously considered.

### Funding
This study was supported by the National Basic Research Program of China (Grant No. 2013CB835100) and the National Natural Science Foundation of China (Grant No. 81570376, No. 31401142 and No. 31401137). The funders had no role in study design, data collection and analysis, decision to publish, or preparation of the manuscript.

### Grant Disclosures
The following grant information was disclosed by the authors:
National Basic Research Program of China: 2013CB835100.
National Natural Science Foundation of China: 81570376, 31401142, 31401137.

### Competing Interests
The authors declare there are no competing interests.

## Author Contributions

- Wen-Xing Li and Shao-Xing Dai conceived and designed the experiments, performed the experiments, analyzed the data, contributed reagents/materials/analysis tools, wrote the paper, prepared figures and/or tables, reviewed drafts of the paper.
- Qian Wang performed the experiments, analyzed the data, prepared figures and/or tables, reviewed drafts of the paper.
- Yi-Cheng Guo and Yi Hong performed the experiments, analyzed the data, reviewed drafts of the paper.
- Jun-Juan Zheng and Jia-Qian Liu prepared figures and/or tables, reviewed drafts of the paper.
- Dahai Liu analyzed the data, reviewed drafts of the paper.
- Gong-Hua Li conceived and designed the experiments, analyzed the data, contributed reagents/materials/analysis tools, wrote the paper, prepared figures and/or tables, reviewed drafts of the paper.
- Jing-Fei Huang conceived and designed the experiments, contributed reagents/materials/analysis tools, reviewed drafts of the paper.

## Microarray Data Deposition

The following information was supplied regarding the deposition of microarray data:
NCBI-GEO GSE16561, GSE22255 and GSE37587.

## Data Availability

The research in this article did not generate, collect or analyse any raw data or code.

## Supplemental Information

Supplemental information for this article can be found online at http://dx.doi.org/10.7717/peerj.2470#supplemental-information.

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
