# Peer review of "Integrated analysis of ischemic stroke datasets revealed sex and age difference in anti-stroke targets"

_PeerJ, doi:10.7717/peerj.2470_

## Round 0.1 · original submission · Major Revisions

In particular I ask that you seriously consider the comments of reviewers 1 and 3 as if you can address their concerns, your manuscript will be much stronger.

Reviewer 1 ·

Basic reporting

a. There are many spelling and grammatical errors throughout the manuscript that would need to be corrected for easier reading and interpretation.
b. The authors have clearly made an effort to provide a thorough introduction and background for the study.
c. The structure is maintained for the most part, but custom scripts that have been used for a key step in the data analysis should be shared.
d. The figures are acceptable.

Experimental design

a. The authors have used publicly available data, but not provided any links to the datasets on the NCBI GEO website (only GSE ID has been provided)
b. The authors have raised an important questions and tried to fill in a well-identified research gap, but need to expand the sample size.
c. Methods are described with sufficient detail.

Validity of the findings

a. This study is, at best, incremental research on a relatively small sample set.
b. At the very least, it would add tremendous value to investigate the potential biomarker genes and pathways in an independent validation cohort of ischemic stroke patients.
c. Overall, there are a few potentially interesting conclusions that would need to be validated.

Additional comments

Good work on coming up with a relevant research question and setting up this study. Demonstrating validity of findings in an independent patient cohort would help increase the value of the study and make it acceptable for publication. Thanks.

·

Basic reporting

The authors report to show sex- and age-dependent gene expression changes in human studies of ischemic stroke by integrated analysis of GEO datasets. They present that biological pathways associated with immune responses were differently enriched between sex and ages. Interesting point is that these pathways were more seriously affected by old female patients than male patients, whereas non-old subjects showed that female were less affected. Another remarkable point is to compare anti-stoke target genes between sex and ages, which indicated two key genes (ARG1 and MMP9) in all groups and higher anti-stroke target genes in comparison of male vs female patients. Given that association of sex and ages in ischemic stroke, these data are intriguing to those interested in sex-biased pathological mechanisms and also gives important insights that sex difference should be seriously considered in the drug development and clinical therapy in this disease.

Overall manuscript is written well with sufficient introduction and background of the sex- and age-related issues in ischemic stroke. All figures are relevant to the content of the article with adequate resolution, and appropriately described and labeled.

Experimental design

No Comments

Validity of the findings

The data are robust, statistically acceptable and reasonable to derive the conclusions and the significance.

Additional comments

The authors should be congratulated on an interesting findings that give important insights on sex-biased mechanisms in ischemic stroke studies. Based on gene expression analysis data, it is suggested that interferon signaling may be an important factor in sex-biased mechanisms. It would be interesting if you try to find the connection of anti-stroke genes to the interferon signaling.

Reviewer 3 ·

Basic reporting

Please consider having this manuscript edited to improve the quality and clarity of English language writing. At times the improper phrasing and syntax make it difficult to understand the science.
• “non-old” is not a word or phrase in the English language. Consider changing to “young”.
• When referring to data in the results section, please use past tense. In the discussion section and introduction the present tense is usually appropriate.
• Please be sure to include the proper article (a/an/the) where appropriate.
• In the results section, please refer to the figures when discussing them; for example, please reference figures 1A and 1B, 2A and 2B, 3A and 3B.

Experimental design

• The authors should highlight the novel findings of their work, as sex and age differences have been reported extensively in the literature. For example, is the finding that ARG1 and MMP9 are upregulated a novel discovery? Or just the methods?
• For Table 1 and Figure 3, please discuss the purpose of showing “total group”, particularly if the focus is on differences between male vs. females and young vs. old.

Validity of the findings

• If the only novel aspect of this manuscript is the methods (integrated microarray data), then the authors should include a discussion for why this method was chosen to replicated already known data. How do the results add to the existing body of literature on sex and age differences in stroke patients?

---

## Round 0.2 · accepted · Accept

I would add that one reviewer still feels more English editing is needed so if the manuscript could be checked once again, that would be appreciated.

Reviewer 1 ·

Basic reporting

The use of a Native English speaker to review and edit the manuscript seems to have worked, and now presents as a cohesive study.

Experimental design

The integrative analysis approach, though not novel in itself, has been applied here with a good approach and context that addresses a certain knowledge gap that exists in this field.

Validity of the findings

Analyzing an independent dataset and validating some of the initial findings makes this study quite robust and reveals interesting candidates that could be developed as potential biomarkers for stroke.

Additional comments

Thank you for addressing all the concerns and comments. The team should be commended for contributing this information and expanding the knowledge about this important disease.

Reviewer 3 ·

Basic reporting

The English hasn't really been improved, so hopefully the editors of PeerJ can help with that aspect.

Experimental design

No further comments

Validity of the findings

No further comments